# KNOWLEDGE MANIPULATION IN LANGUAGE MODELS (PART B)*

## ABSTRACT

Language models can store vast amounts of factual knowledge, but their ability to use this knowledge for logical reasoning remains questionable. This paper explores a language model's ability to manipulate its stored knowledge during inference. We focus on four manipulation types: **retrieval** (e.g., "What is person A's attribute X"), **classification** (e.g., "Is A's attribute X even or odd?"), **comparison** (e.g., "Is A greater than B in attribute X?") and **inverse search** (e.g., "Which person's attribute X equals T?") We observe that pre-trained language models like GPT2/3/4 excel in knowledge retrieval but struggle with simple classification or comparison tasks unless Chain of Thoughts (CoTs) are employed during both training and inference. They also perform poorly in inverse knowledge search, irrespective of the prompts. Our primary contribution is a synthetic dataset for a *controlled experiment* that confirms these inherent weaknesses: a language model cannot *efficiently* manipulate knowledge from pre-training data, even when such knowledge is perfectly stored and fully extractable in the models, and despite adequate instruct fine-tuning.

## 1 INTRODUCTION

Knowledge is a fundamental component of human civilization and intelligence. Throughout our lives, we accumulate a vast amount of knowledge and learn to use it flexibly. Recently, large language models like GPT4 (OpenAI, 2023) have demonstrated an impressive capacity to memorize extensive amounts of knowledge, arguably more than any human can. These models also show signs of being able to manipulate this knowledge to solve various problems.

In this work, we aim to understand how **transformer based language models** manipulate the knowledge they have memorized during pretraining and use it flexibly to solve different tasks at inference time. For instance, can the language model answer questions like "Is Joe Biden older than Donald Trump" based on its memorization of the two presidents' birthdays? Can it infer whether Princeton is ranked higher than MIT based on its stored 2023 US News university ranking knowledge?

In this paper, we consider a language model's ability to answer questions during inference time, where those questions are some *functions* of specific knowledge in its pretraining. These questions or their equivalent forms may not be in the model's training data, but *the same function* for other knowledge should have been (so the model understands the function). For instance, can the model answer "Was Joe Biden born in an even year?" if it *hasn't encountered this sentence or its equivalents during pretraining* (such as "Is Joe Biden's birth year divisible by 2"), but inferring from "Biden was born in 1942" and "1942 is even"? Answering such questions necessitates the model to both memorize and comprehend the knowledge.

**Knowledge manipulation is a form of logical reasoning**. To answer questions like "Is Person A's attribute X Good?", a language model not previously exposed to this sentence in its pretraining data may draw from other training data such as "person A's attribute X equals T" and "T is Good".

In this paper, "knowledge" refers to *factual knowledge* (e.g., knowledge graph), and we examine if a language model can logically manipulate such knowledge stored in the model weights. Other studies may investigate in-context knowledge or RAG (Cai et al., 2022; Jiang et al., 2023; Komeili

---

*Since "knowledge" is a broad subject, we have to write separate papers to cover its different aspects. Our Part A (Anonymous, 2023) addresses how knowledge is *stored*, the conditions under which knowledge can be *extracted* through instruct fine-tuning, and introduces probing techniques. This Part B is built on it to study how such knowledge can be *further manipulated* for downstream tasks. We've anonymously submitted both Part A and B to ICLR 2024 as standalone papers, ensuring no result overlap and making each self-contained. Our Part A is also in the supplementary material for interested readers.

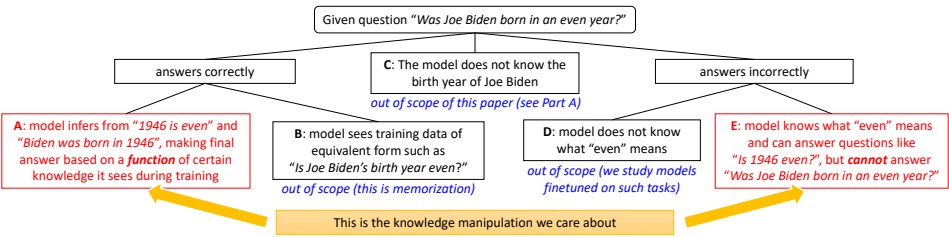

Figure 1: We study (A) vs (E) as knowledge manipulation. With a pre-trained model over internet data, it is very hard to determine whether (B,C,D) has happened due to the uncontrollability of internet data.

et al., 2021; Lewis et al., 2020; Liu et al., 2020; Mao et al., 2020; Parvez et al., 2021; Ram et al., 2023; Siriwardhana et al., 2023), where the model, given a paragraph during inference, immediately answers logic questions about it.

Research has extensively explored language models' question-answering abilities during inference time (Hernandez et al., 2023; Naseem et al., 2021; Omar et al., 2023; Peng et al., 2022; Petroni et al., 2019; Richardson and Sabharwal, 2020; Singhal et al., 2022; Sun et al., 2023). However, these studies primarily focus on models trained on internet data. A key challenge in understanding whether these models can manipulate knowledge is discerning whether the internet data already includes the exact or equivalent sentence, or if the models have correctly stored such knowledge and retrieved it from inference time. Refer to Figure 1.

To address the *unpredictability of internet data*, our concurrent study (Anonymous, 2023) created synthetic pretraining data containing a controlled biography of $N = 100k$ individuals and pretrained a language model on this data. This prior work investigates *when and how* the model can store and retrieve knowledge about these 100k individuals during inference time after pretraining. Here is an example of the biography data:

Anya Briar Forger was born on October 2, 1996. She spent her early years in Princeton, NJ. She received mentorship and guidance from faculty members at Massachusetts Institute of Technology. She completed her education with a focus on Communications. She had a professional role at Meta Platforms. She was employed in Menlo Park, CA.

$$(1.1)$$

Our concurrent work (Anonymous, 2023) suggests that a pretrained model may struggle to extract stored knowledge from biographical data unless the data is sufficiently *augmented*, such as when English sentences exhibit ample diversity and are sufficiently permuted (see Section 2). This facilitates knowledge extraction for questions like "Which city was Anya Briar Forger born in?" While we recommend reading our concurrent work first, this paper can be read independently.

This paper further explores whether the model, pre-trained on the already augmented biography data, can manipulate its stored (and extractable) knowledge after instruction finetune. For instance, can it answer questions requiring reasoning based on a person's attributes, such as "Was Anya born in a southern city?" or "Is Anya's university better than Sabrina's?" using information from their biographies. During training, we expose the model to the biographies of all $N$ people and the knowledge manipulation question-answer (QA) texts from a subset of individuals (the in-distribution set $\mathcal{P}_{\text{train}}$). We test the model's *out-of-distribution* generation accuracy on correctly answering such manipulation questions for the remaining subset (the out-of-distribution set $\mathcal{P}_{\text{test}}$), where the model sees the biographies during training, but not the QAs. The existence of the set $\mathcal{P}_{\text{train}}$ in the training data ensures that the model is exposed to enough training examples to understand the QAs.

## 1.1 OVERVIEW OF THE KNOWLEDGE MANIPULATION TASKS

Our paper explores four types of knowledge manipulation: retrieval, classification, comparison, and inverse search, which we believe encompass most real-world knowledge manipulation scenarios. Each manipulation task is discussed separately below.

RETRIEVAL. We extend our work on knowledge extraction (Anonymous, 2023) to a more general setting. After pretraining on biographical data, we finetune the model to (1) retrieve part of an attribute or (2) retrieve multiple attributes simultaneously. We find that retrieving multiple attributes is nearly as easy as extracting individual ones. However, in this broader context, partial retrieval can be significantly more challenging. For instance, a language model may correctly answer "What is the birth date of Anya" as "June 27th, 1997", but struggle with "What is the birth year of Anya".

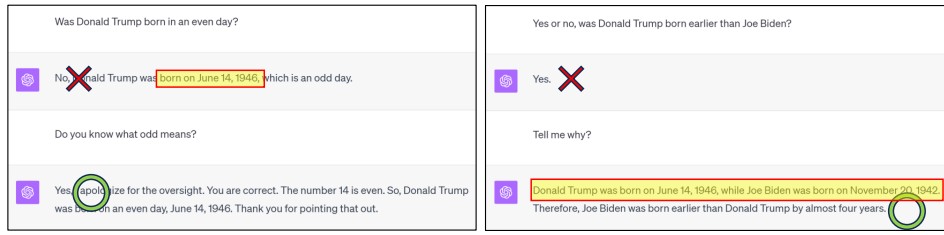

Figure 2: GPT4 struggles to answer simple manipulation questions based on a person's attributes during inference, despite knowing the knowledge. When a Chain of Thoughts (CoT) approach is used, where the person's attributes are first explicitly spelled out, GPT4 can correctly answer the manipulation tasks. More ChatGPT examples and details are in Figure 5, Figure 7, Figure 10, and Appendix E.

CLASSIFICATION. Classification tasks involve determining the validity of a statement based on individual attributes. For instance, answering "What degree did Anya receive?" requires a ternary response (art, science, engineering) based on her major of study. After training on the biography data, we find that language models often struggle with such tasks unless they generate answers in a Chain of Thought (CoT) manner or are finetuned with a much larger number of samples of such tasks than information theoretically necessary. For example, a model might correctly answer "What is Anya's birth month," but fail to determine if it's even without first generating the birth month and then assessing its parity. This remains true even after the model can answer one's birth month with nearly 100% accuracy, and further trained/finetuned with 25,000 individuals on birth-month parity questions (both with direct answers like "Alice's birth month is even" and with Chain of Thoughts like "Alice's birth month is 12, 12 is even"), far exceeding the number needed to classify 12 months into 2 classes. Our findings suggest that the model **can not be trained/finetuned efficiently to perform even a single step of basic knowledge manipulation** during inference time without using CoT, even if it sees a lot of such single-step (either non-CoT or CoT!) knowledge manipulation training data.

COMPARISON. Comparison involves determining whether one attribute is greater or smaller than another, based on a predefined ranking or order. For example, "Is Anya's university better than Sabrina's?" requires a "Yes" or "No" response based on the universities and their ranking. Similar to classification, we find that language models **cannot be trained/finetuned efficiently to perform this type of knowledge manipulation** unless they generate answers in a CoT manner.

INVERSE SEARCH. This involves identifying a person based on their attributes, such as "Who was born in 1996 in Princeton, NJ?" or "Who studied Communications at MIT and worked for Meta?". We find that language models **cannot perform this task, regardless of training methods or the volume of training examples**, unless the knowledge is presented inversely in the training data, such as "Born in 1991 in New York was Anya" or "Studied at MIT and worked for Meta was Anya". Merely having forward knowledge data like "Anya was born in 1996 in Princeton, NJ" or "Anya studied at MIT and worked for Meta" in the training is unsufficient, irrespective of augmentation or finetuning strategies. A concurrent study (Berglund et al., 2023) also observed a similar "reversal curse". This strongly suggests that **language models cannot function as databases**.

**Our contribution.** We demonstrate that pre-trained language models, using synthetic biography data, perform poorly at knowledge manipulation. Regardless of the pretraining/finetuning, they still struggle with simple functions about a person's attributes, such as "Is person A's birth month even?" unless the function of *the same person* is in the training data. This can be mitigated by training/prompting the model to answer in a Chain of Thought (CoT) manner. However, the model fails at inverse knowledge search, *regardless of prompting/training*. It can generate all attributes of a person given the person's name, but not vice versa. Even large models like GPT-4 (see Figure 2) perform poorly at these tasks, suggesting **these limitations may be inherent to generative language models and not resolved by scaling up**, but require novel techniques to improve the model's knowledge manipulation ability. Our synthetic setting serves as a *simple, yet important testbed* for future work on enhancing language models' knowledge manipulation abilities.

## 2 PRELIMINARIES

To ensure that this paper is self-contained, we briefly summarize some of the datasets, terminologies, models, and training methods introduced in Anonymous (2023).

**BIO datasets bioS.** In Anonymous (2023), we presented a synthetic biography (BIO) dataset, bioS, consisting of $N = 100,000$ individuals with six attributes: birth date, birth city, university, major, employer, and working city.[1] Six randomly chosen English sentences describe each individual's attributes as in (1.1). This basic setup, that we call "bioS single", has one biographical entry per individual with six sentences in the order of (1.1).

We also explored *knowledge augmentation* in Anonymous (2023), including: (1) multi$M$, generating $M$ equivalent entries per person (using different wordings); (2) permute, random sentence shuffling; and (3) fullname, replacing pronouns with full names. We considered 15 augmentations, combinations of the above. For instance, "bioS multi5+permute" denotes five biographical entries per individual with shuffled sentences. (Refer to Figure 3 or Appendix A for a complete list of such augmentations.)

**BIO dataset bioR.** We also introduced a realistic bioR dataset in Anonymous (2023), created using LLaMA-30B (Touvron et al., 2023; Zhou et al., 2023) to write entries similar to real biographies. This paper uses bioS for negative results and both bioS and bioR for positive results.

**QA dataset and single knowledge extraction.** In Anonymous (2023), we analyzed QAs like "What is the birth city of Anya Briar Forger?" corresponding to six individual attributes. We split the $N$ individuals into two equal parts: a training set $\mathcal{P}_{\text{train}}$ and a testing set $\mathcal{P}_{\text{test}}$. We then explored two training methods:

- In *BIO+QA mixed training*, we simultaneously trained the language model on the BIO for everyone and QA data for $\mathcal{P}_{\text{train}}$, using a *high* ratio QA$_r$ to control the percentage of QA data.
- In *BIO pretrain + QA finetune*, we initially pretrained the language model with the BIO data, then fine-tuned it using the QAs for individuals in $\mathcal{P}_{\text{train}}$.

In both cases, we assessed the model's accuracy to answer questions about individuals in $\mathcal{P}_{\text{test}}$, referred to as *QA test accuracy*. **Key findings** from our parallel paper Anonymous (2023) include:

- The success of QA finetune largely depends on pretraining data *augmentation*. For instance, pretraining on bioS multi5+permute yields a mean knowledge extraction accuracy over $96.6\%$, while bioS single results in just $9.7\%$ accuracy (see right block of Figure 3).[2]
- In BIO+QA mixed training, knowledge augmentation is less critical, with the model achieving over $85\%$ QA test accuracy on bioS single. However, as shown in (Anonymous, 2023), this method mirrors a "study to pass the test" approach, where the knowledge is first learned from QAs, unlike typical human knowledge acquisition and is also less practical.

**Language models.** The standard GPT2-small architecture comprises 12 layers with 12 heads and 768 dimensions (Radford et al., 2019). However, its performance can be limited by its absolute positional embedding. Thus, we use its rotary positional embedding variant (Black et al., 2022; Su et al., 2021), still referred to as GPT2 for short. We train GPT2-small on bioS, but use a larger 12-layer, 20-head, 1280-dim GPT2 for bioR to handle the increased data complexity. A fixed context window length of 512 is used throughout this paper.

## 3 WARM-UP ON KNOWLEDGE RETRIEVAL

We examine two *partial knowledge retrieval* tasks that involve extracting either the person's birth day or year from the complete birth date information.

1. What is the birth day of Anya Briar Forger? *2*.      2. What is the birth year of Anya Briar Forger? *1996*.

We consider six *dual knowledge retrieval* tasks:

1. Where was Anya Briar Forger born and which company did this this person work for? *Princeton, NJ; Meta Platforms.*
2. Which company did Anya Briar Forger work for and where was this person born? *Meta Platforms; Princeton, NJ.*
3. Which university and what major did Anya Briar Forger study? *Massachusetts Institute of Technology; Communications.*
4. What major and which university did Anya Briar Forger study? *Communications; Massachusetts Institute of Technology.*
5. Where and which company did Anya Briar Forger work for? *Menlo Park, CA; Meta Platforms.*
6. Which company and where did Anya Briar Forger work for? *Meta Platforms; Menlo Park, CA.*

---

[1] All fields, except the working city (determined by the employer's headquarters), are randomly selected.

[2] In Anonymous (2023), we used probing to explain this phenomenon. Essentially, knowledge augmentation in the BIO pretraining data ensures that knowledge is more closely tied to an individual's name.

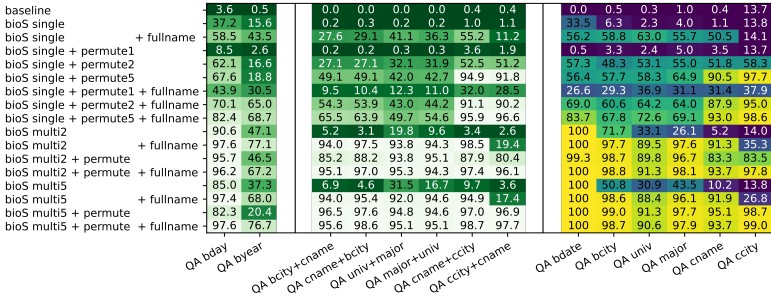

Figure 3: Partial (left) and dual (middle) knowledge retrieval, versus the single knowledge extraction (right).

> **Each row** denotes a unique pretrained model with its corresponding knowledge augmentation on the bioS data. The **left**, **middle**, and **right blocks** depict QA finetune test accuracies for *partial*, *dual*, and *single knowledge retrieval* tasks, with the right block from Anonymous (2023). Details on the knowledge augmentations and additional experiments on the bioR dataset are in Appendix A and B.

**Methodology.** We aim to determine if a model pretrained on BIO data can be fine-tuned to address the eight questions related to partial or dual knowledge retrieval. We divide the $N$ individuals equally into training set $\mathcal{P}_{\text{train}}$ and testing set $\mathcal{P}_{\text{test}}$. The model is fine-tuned using the above eights QA tasks for individuals in $\mathcal{P}_{\text{train}}$ and evaluated on its *out-of-distribution* generation accuracy by testing its responses to the questions for individuals in $\mathcal{P}_{\text{test}}$. We use LoRA fine-tuning Hu et al. (2021) to enhance performance, as suggested by Anonymous (2023) (see Appendix B for details).

**Our findings.** For dual knowledge retrieval, the fine-tuning accuracy largely depends on the extractability of knowledge related to the two individual tasks from the retrained model.

- If a language model is pretrained on sufficiently augmented data, such as bioS multi5+permute, which generates five biographical entries per person and permutes the six sentences randomly, the accuracy for dual knowledge retrieval is nearly perfect. In other words, **dual knowledge retrieval is straightforward when the individual tasks are**.
- However, if the pretraining data exhibits spatial dependency between the two knowledge pieces, the *sequence of their retrieval can impact accuracy*. For example, with bioS multi5+fullname, where biographical entries always maintain the same order (specifically, the company name always precedes the company city, and recall company city is uniquely determined by the company name as noted in Footnote 1), answering the company name first yields near-perfect accuracy, but answering the company city first drastically reduces accuracy.

Even with near-perfect extraction of an attribute (e.g., a birth date), partial retrieval (e.g., the birth year) may still be poor. The model may fail to answer questions like "What is the birth *year* of person Anya", despite correctly answering "What is the birth date of person Anya".

- This is preliminary evidence that the model requires Chain of Thoughts (CoTs) for knowledge manipulation. For instance, during inference, the model must state the birth month before the birth year, adhering to the data order in pretraining. It may not necessarily be able to "skip" tokens to directly generate subsequent knowledge from pretraining.

## 4 OUR RESULTS ON KNOWLEDGE CLASSIFICATION AND COMPARISON

This section illustrates that a generative model, despite its ability to extract knowledge effectively, may struggle with downstream tasks requiring basic operations to manipulate such knowledge. This is unless the Chain of Thought (CoT) is implemented during *both* training and testing phases.

**Classification QA.** We investigate classification tasks related to a person's birth month and field of study. For the birth month, we use modular arithmetic with $p = 2, 6, 12$:

1. Was Anya Briar Forger born in an even month? Answer: *Yes*.
2. What is Anya Briar Forger's birth month mod 6? Answer: *4*.
3. What is Anya Briar Forger's birth month in numerics? Answer: *10*.

We assigned a "luckiness" index to 100 unique majors in our BIO dataset.[3] We then queried "What is the luckiness of Anya Briar Forger's major modulo $m$?" for $m = 5, 20, 100$. Classifying birth

---

[3] For instance, Computer Science is 0, Communications is 28, and Music is 99.

| field | task | #train individuals | baseline | pretrained model | | | | QA finetuned model | | | |
|---|---|---|---|---|---|---|---|---|---|---|---|
| | | | | trained w/o hint | trained with hint | | | trained w/o hint | trained with hint | | |
| | | | | test acc | test acc (with hint) | test acc (w/o hint) | hint acc | test acc | test acc (with hint) | test acc (w/o hint) | hint acc |
| birthmonth | classify %2 | (2.5k) | 50.0 | 60.4 | 77.8 | 65.2 | 64.5 | 61.9 | 80.4 | 65.2 | 69.1 |
| birthmonth | classify %2 | (5k) | 50.0 | 67.3 | 87.3 | 72.7 | 80.3 | 68.0 | 89.5 | 72.8 | 83.9 |
| birthmonth | classify %2 | (10k) | 50.0 | 75.9 | 94.2 | 80.3 | 91.0 | 76.4 | 95.0 | 79.9 | 92.8 |
| birthmonth | classify %2 | (25k) | 50.0 | 86.4 | 98.6 | 91.1 | 97.8 | 87.1 | 98.8 | 90.9 | 98.4 |
| birthmonth | classify %2 | (50k) | 50.0 | 95.3 | 99.5 | 97.5 | 99.2 | 96.3 | 99.7 | 97.5 | 99.5 |
| birthmonth | classify %12 | (2.5k) | 8.3 | 51.5 | 61.5 | 53.7 | 61.5 | 58.3 | 64.1 | 53.8 | 64.0 |
| birthmonth | classify %12 | (5k) | 8.3 | 74.2 | 79.0 | 70.1 | 79.0 | 80.3 | 82.5 | 75.0 | 82.4 |
| birthmonth | classify %12 | (10k) | 8.3 | 91.6 | 92.0 | 86.8 | 92.0 | 93.5 | 94.7 | 91.2 | 94.7 |
| birthmonth | classify %12 | (25k) | 8.3 | 97.9 | 98.5 | 96.8 | 98.5 | 98.9 | 99.2 | 98.3 | 99.2 |
| birthmonth | classify %12 | (50k) | 8.3 | 99.4 | 99.5 | 99.4 | 99.5 | 99.6 | 99.8 | 99.7 | 99.8 |
| birthmonth | ranking | (2.5k) | 54.2 | 53.7 | 65.4 | 59.6 | 44.2 | 57.3 | 65.5 | 57.6 | 44.9 |
| birthmonth | ranking | (5k) | 54.2 | 59.2 | 75.5 | 63.4 | 63.6 | 62.5 | 75.1 | 63.1 | 62.6 |
| birthmonth | ranking | (10k) | 54.2 | 65.4 | 87.7 | 67.0 | 82.7 | 65.9 | 88.9 | 66.3 | 83.9 |
| birthmonth | ranking | (25k) | 54.2 | 75.6 | 96.7 | 75.8 | 95.4 | 78.3 | 97.4 | 72.5 | 96.3 |
| birthmonth | ranking | (50k) | 54.2 | 85.6 | 99.0 | 86.7 | 98.5 | 88.6 | 98.9 | 82.9 | 98.3 |
| major | classify %5 | (10k) | 20.0 | 23.6 | 86.4 | 24.1 | 84.5 | 22.8 | 89.6 | 23.9 | 87.9 |
| major | classify %5 | (25k) | 20.0 | 24.6 | 96.7 | 26.8 | 96.3 | 24.8 | 97.7 | 27.0 | 97.2 |
| major | classify %5 | (50k) | 20.0 | 31.6 | 99.3 | 34.2 | 99.2 | 30.0 | 99.5 | 33.9 | 99.4 |
| major | classify %100 | (10k) | 1.0 | 30.1 | 78.7 | 34.6 | 79.0 | 8.9 | 75.8 | 22.2 | 76.1 |
| major | classify %100 | (25k) | 1.0 | 79.3 | 96.0 | 74.4 | 96.0 | 80.0 | 95.6 | 77.1 | 95.3 |
| major | classify %100 | (50k) | 1.0 | 91.7 | 99.0 | 90.7 | 99.1 | 91.8 | 98.3 | 92.5 | 98.1 |
| major | ranking | (10k) | 50.5 | 52.5 | 88.8 | 54.1 | 86.2 | 52.4 | 90.3 | 54.1 | 88.3 |
| major | ranking | (25k) | 50.5 | 52.2 | 96.4 | 53.7 | 97.3 | 52.6 | 96.9 | 53.6 | 97.5 |
| major | ranking | (50k) | 50.5 | 53.9 | 99.6 | 55.0 | 99.5 | 53.6 | 99.4 | 55.0 | 99.3 |
| major | subtraction | (10k) | 1.0 | 1.1 | 21.6 | 1.1 | 82.5 | 1.0 | 23.2 | 1.1 | 84.3 |
| major | subtraction | (25k) | 1.0 | 1.1 | 89.1 | 1.2 | 96.7 | 1.2 | 84.7 | 1.2 | 97.0 |
| major | subtraction | (50k) | 1.0 | 1.1 | 98.4 | 1.2 | 99.3 | 1.1 | 97.3 | 1.2 | 99.0 |

Figure 4: Knowledge classification and comparison tasks on a BIO pretrained model vs a QA finetuned model. The **#train individuals** column shows the size $|\mathcal{P}_{\text{train}}|$. The **trained w/o hint** column indicates the model finetuned on the classification/comparison tasks without adding hints. The **trained with hint** block shows the model finetuned with hints added with a probability of 0.5. **Test acc (with hint)** and **test acc (w/o hint)** represent the accuracy for individuals in $\mathcal{P}_{\text{test}}$ with or without hints, while **hint acc** shows the model's hint generation accuracy. See Figure 9 and Appendix C for more experiments.

month with $p = 12$ or major with $p = 100$ is a form of *transfer learning*, similar to tasks in (Anonymous, 2023), but with altered question phrasing and response format.

**Knowledge comparison QA.** We examine tasks related to *ranking* and *subtraction* based on a person's birth month and major of study. The questions include:[4]

1. Was Anya Briar Forger born in a month in a year later than Sabrina Eugeo Zuberg? [Yes/No].
2. What is Anya Briar Forger's birth month minus Sabrina Eugeo Zuberg's birth month? [-11..11].
3. Did Anya Briar Forger major in a field luckier than Sabrina Eugeo Zuberg? [Yes/No].
4. How luckier is Anya Briar Forger's major compared with Sabrina Eugeo Zuberg's major? [-99..99]

**Methodology.** We evaluate knowledge manipulation using a model proficient in knowledge extraction, ensuring any difficulties arise from manipulation, not extraction. We utilize our knowledge-augmented biographical data, bioS multi5+permute, which allows nearly $100\%$ test accuracy for extracting birth date/month and 97.9% for major of study.

We employ either a model pretrained from this BIO data (the *BIO pretrained model*), or one that is BIO pretrained + QA finetuned for single knowledge extraction tasks, such as "What is the birth date of Anya Briar Forger?" (the *QA finetuned model*). Given the QA finetuned model's proven extraction ability, one might anticipate a better performance in knowledge manipulation.

TRAIN WITHOUT HINT. Our BIO data comprises biographical entries of $N = 100k$ individuals. We reserve half (i.e., $50k$) as the testing set $\mathcal{P}_{\text{test}}$, and select a separate subset $\mathcal{P}_{\text{train}}$ as the training set, with $|\mathcal{P}_{\text{train}}| = 2.5k, 5k, \ldots 50k$.

Starting from one of the two aforementioned models, we perform additional LoRA fine-tuning using the classification or comparison QA tasks described earlier, trained using individuals from $\mathcal{P}_{\text{train}}$. We then evaluate the model's *out-of-distribution* generation accuracy by assessing its performance on the classification/comparison tasks for individuals in $\mathcal{P}_{\text{test}}$.

TRAIN WITH HINT. To enhance the model's knowledge manipulation capabilities, we LoRA fine-tune it using *knowledge hints*. These hints state a person's attributes in English before answering the manipulation question. For example, in our tasks, the underlined sentences serve as hints:[5]

1. Was Anya Briar Forger born in a month in a year later than Sabrina Eugeo Zuberg? October; September. No.
2. How luckier is Anya Briar Forger's major compared with Sabrina Eugeo Zuberg's major? Communications; Music. -71.
3. What is the luckiness of Anya Briar Forger's major modular 20? Communications. 8.

---

[4]These questions have practical relevance: our luckiness index could be replaced with, for instance, the popularity of majors from US News.

[5]For context, apart from (1.1), we consider another individual Sabrina Eugeo Zuberg who was born in September and majored in Music. We have previously assigned specific luckiness values to each major: Communications holds a value of 28, while Music is valued at 99.

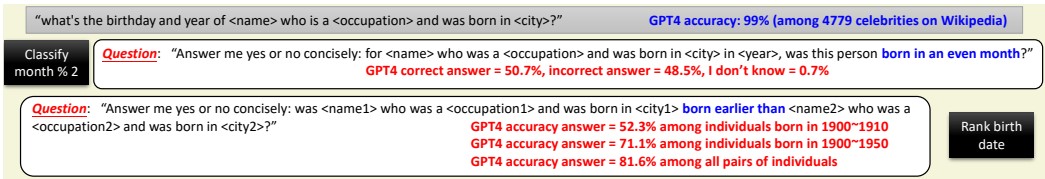

Figure 5: Knowledge classification and ranking on WikiBio using GPT3.5 / GPT4. Details are in Appendix E.2.

Including hints in the training data allows the language model to use a *chain of thought (CoT)* approach: it can first extract the necessary knowledge and then learn the manipulation task by directly using this knowledge. Similar to "train without hint", we train using QAs for individuals in $\mathcal{P}_{\text{train}}$ and test on $\mathcal{P}_{\text{test}}$. For each individual in $\mathcal{P}_{\text{train}}$ (or each pair for comparison tasks), we include hints with a 50% probability. Thus, the model sees training data *both with and without hints*. We then test the model's *out-of-distribution* generation accuracy under both conditions. Our goal is to determine: does the **integration of CoT data improve the model's knowledge manipulation skills, even without CoT?**

**Our Findings.** As shown in Figure 4, we found significant challenges in knowledge classification/comparison unless hints are used consistently throughout training and testing. Specifically, we observed:

1. The difference between a BIO pretrained and a QA finetuned model is minimal for downstream knowledge manipulation tasks. Fine-tuning the model to answer questions like "What major did Anya Briar Forger study" does not necessarily improve its performance on future tasks like ranking and classification based on the major of study.

2. Without CoT examples, the model's test accuracy is significantly low, even for simple tasks.
   - Determining whether a month is even or odd requires 10,000 training samples to achieve a 75% accuracy, despite theoretically needing a sample complexity on the order of $O(12)$.
   - Ranking months requires $50,000$ training samples to reach an $85\%$ test accuracy, even with a theoretical sample complexity of $O(12^2)$, provided no hint is given.
   - The "transfer learning" task, which involves rephrasing the same knowledge, has a relatively better test accuracy.
   - Classifying or ranking majors from a list of 100 possible majors barely outperforms random guessing, even with a maximum of $50,000$ training individuals.

3. When CoT examples are included during training:
   - The model still struggles to answer without a hint during testing, indicating that *including hints during training does not improve test-time accuracy when hints are removed*.
   - However, when the model is prompted with a hint during testing, there's a significant improvement in test accuracy, closely aligning with the accuracy achieved when producing the intermediate steps. For example:
     - In the task "birth month classify %2", with a hint accuracy 91.0%, the test accuracy (with hint) is 94.2%, nearly aligning with the calculation: $91.0\% + (1-91.0\%) \times 50\% = 95.5\%$.
     - In the task "birth month subtraction", a hint accuracy of 78.1% results in a test accuracy (with hint) of 61.5%, comparable to the value derived from the formula: $78.1\% \times 78.1\% + (1 - 78.1\% \times 78.1\%) \times 8.3\% = 64.2\%$.
   
   Thus, in scenarios with CoTs, if the model can accurately navigate the intermediate step, it is highly likely to successfully tackle the subsequent manipulation task, and vice versa.

**Connection to GPT4 in practice.** Figure 5 illustrates GPT4's struggle with biographical data classification and comparison tasks in the absence of CoTs. Figure 2 and Figure 10 show that CoTs can rectify this. This suggests that scaling up model size may not mitigate the issues discovered in this section. The GPT4 experiment is included solely for illustrative purposes. Without control over its pretrained data, distinguishing between Case (A)-(E) from Figure 1 is difficult. In Figure 5, we ensured the model could accurately identify individuals' birth dates 99% of the time, thereby eliminating Case (C). However, we cannot dismiss Case (D) due to uncertainty about the number of relevant training examples in GPT4's data. Interestingly, GPT4 has a 71.1% accuracy rate when comparing birth dates from 1900-1950, but this drops to 52.3% for 1900-1910, suggesting a correlation with the number of samples in its training data. Therefore, our primary focus of this paper is on a controlled, synthetic experiment to study knowledge manipulation.

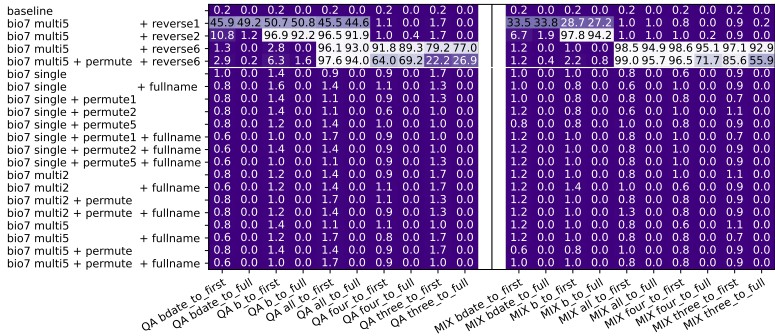

Figure 6: Test accuracy for QA finetune (left) and BIO+QA mixed-training (right) in knowledge inverse search. **Each row** denotes a unique pretrained model with its corresponding knowledge augmentation on the bioS data. The 4 rows with reverse indicate knowledge written in reverse on the pre-training data for comparison (thus, these rows are no longer *inverse* search). Refer to Appendix D for more details.

## 5 OUR RESULTS ON KNOWLEDGE INVERSE SEARCH

We now show that a generative model cannot typically perform a knowledge inverse search, **unless the knowledge was already pretrained in reverse order**.

**Knowledge inverse search.** Remember that our biographical entry in the bioS data always starts with the person's name, as shown in (1.1). This enables us to examine the knowledge inverse search by asking about the individual's first or full names. We consider 10 such QA tasks (with their task names provided on the right):

- Give me the [first/full] name of the person born on October 2, 1996? (bdate_to_first, bdate_to_full)
- Give me the [first/full] name of the person born on October 2, 1996 in Princeton, NJ? (birth_to_first, birth_to_full)
- Give me the [first/full] name of the person who studied Communications at Massachusetts Institute of Technology and worked for Meta Platforms? (three_to_first, three_to_full)
- Give me the [first/full] name of the person who studied Communications at Massachusetts Institute of Technology, was born in Princeton, NJ, and worked for Meta Platforms? (four_to_first, four_to_full)
- Give me the [first/full] name of the person who studied Communications at Massachusetts Institute of Technology, was born on October 2, 1996 in Princeton, NJ, and worked for Meta Platforms at Menlo Park, CA? (all_to_first, all_to_full)

Note that in our data, some inverse search tasks may not have unique answers (like bdate_to_full). However, with $N = 100,000$ people and $200 \times 12 \times 28$ possible birth dates, a successful inverse search should answer these questions with an accuracy significantly above zero.

**Methodology.** We split the $N$ individuals equally into training set $\mathcal{P}_{\text{train}}$ and testing set $\mathcal{P}_{\text{test}}$. The model is trained using QA data from $\mathcal{P}_{\text{train}}$ and evaluated on its *out-of-distribution generation accuracy* using 10 inverse knowledge search questions for $\mathcal{P}_{\text{test}}$.

We consider two approaches: "BIO pretrain + QA finetune", which fine-tunes a BIO-pretrained model using the above 10 QA tasks on $\mathcal{P}_{\text{train}}$, and "BIO+QA mixed training", where the model is concurrently trained on all the BIO data and 10 QA tasks on $\mathcal{P}_{\text{train}}$. As per Section 2, the latter approach yields better out-of-distribution QA generation accuracies. Details are in Appendix D.

We also introduce new knowledge augmentation variants on the pretraining BIO data for comparison (in addition to the augmentations discussed in Section 2):

- bioS multi5+reverse1, in this case we move the full name of the person to the second sentence:

  The person was born on October 2, 1996. Anya Briar Forger spent her early years in Princeton, NJ...

- bioS multi5+reverse2, in this case we move the full name of the person to the third sentence:

  The person was born on October 2, 1996. She spent her early years in Princeton, NJ. Anya Briar Forger...

- bioS multi5+reverse6, we move the full name of the person to the end of the biographical entry:

  The person was born on October 2, 1996. She spent her early years in Princeton, NJ... The person's name is Anya Briar Forger.

- bioS multi5+permute+reverse6, in this case on top of bioS multi5+reverse6 we also randomly permute the six sentences. Here is an example.

  The person spent her early years in Princeton, NJ. [... 4 more sentences in random order ...] She had a professional role at Meta Platforms. The person's name is Anya Briar Forger.

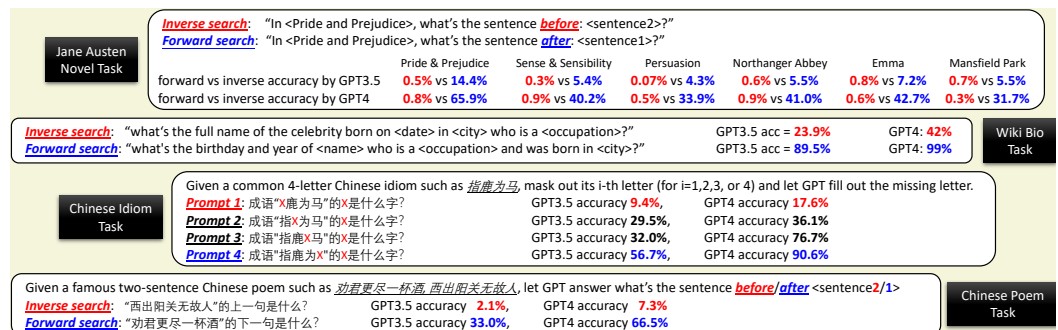

Figure 7: Forward search vs inverse search on ChatGPT (GPT3.5 / GPT4).

**Our findings.** Our results, shown in Figure 6, reveal:

- The model has almost zero accuracy in the inverse knowledge search on the test set $\mathcal{P}_{\text{test}}$, **even with** strong pretraining knowledge augmentation like bioS multi5+permute+fullname,[6] **even for** the simplest inverse knowledge task, all_to_first, and **even with** the BIO+QA mixed training approach.

- However, when the order of knowledge is reversed during pretraining, presenting some attributes before the person's name (see bioS multi5+reverse$P$ or bioS multi5+permute+reverse6), the QA test accuracies improve. This is for illustration purpose only; once the order is reversed, the QA task is no longer an *inverse* knowledge search.

In conclusion, our findings underscore a **fundamental limitation** of the generative model: it cannot perform an inverse knowledge search **unless the knowledge was pretrained in reverse order**. This is due to its left-to-right autoregressive training design. For instance, if the model learns "A equals B" during pretraining, it cannot infer "B equals A" unless it was also in the training data. A bidirectional model like BERT can somewhat mitigate this limitation. However, BERT-like models have their own issues even with forward, single knowledge extraction, even with extensive knowledge augmentation, as discussed in Anonymous (2023).[7]

**Connection to GPT3.5/4 in practice.** Large-scale language models such as GPT3.5/GPT-4 exhibit huge difficulties with inverse knowledge search (Figure 7). For example, while GPT4 can predict the next sentence in Jane Austen's *Pride and Prejudice* with 65.9% accuracy, it only manages 0.8% accuracy when tasked with predicting the preceding sentence. This indicates a deficiency in inverse knowledge search capabilities, regardless of their forward knowledge accuracy and model size.

## 6 CONCLUSION

In this paper, we design a synthetic biography dataset and use it to perform controlled experiments showing the fundamental limitation of the language model's ability to manipulate knowledge during inference time even under the strongest pretraining setting. Our work sheds light on why extremely large language models like GPT4 are still bad at knowledge manipulation, and give surprisingly simple examples (recall "Was Joe Biden born in an even month?") in which Chain of Thought becomes necessary. On the other hand, the language model simply can not perform an inverse search, indicating its limitation to be used as a database. Our synthetic dataset can also be used as an important testbed for designing novel training approaches to mitigate this issue in the future. We believe that our work gives strong support that the language model should be paired with the knowledge base during inference time (retrieval augmented generation (Lewis et al., 2020)) to perform knowledge manipulation efficiently, as it can not be solved efficiently by scaling up the model size/data size.

---

[6]This implies the BIO data includes five diverse biographical entries per individual, with the full name in each sentence, and random shuffling of the six attribute sentences.

[7]As per Anonymous (2023), BERT-like models already struggle with knowledge extraction due to their whole-word masked language modeling (MLM) nature — not to say knowledge manipulation. For example, a company attribute like "Meta Platforms" might lead BERT to correlate the embedding of "Meta" with that of "Platform", rather than associating the company information to the individual's full name. For more details, see our separate paper (Anonymous, 2023).

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
