# OpenReview forum: "Knowledge Manipulation in Language Models (Part B)"
_ICLR.cc/2024/Conference — ICLR 2024 Conference Withdrawn Submission_

### Official Review · Reviewer_kTy8 · 2023-10-23

**Soundness:** 1 poor
**Presentation:** 3 good
**Contribution:** 1 poor
**Rating:** 3
**Confidence:** 5

**Summary:**

Authors try to argue that LLMs are still bad for knowledge manipulation, especially for reasoning and inverse search. The method that the authors used is to carry out four kinds of knowledge manipulation tasks on a GPT2-X model trained using a synthetic biography dataset (GPT2-small for bioS dataset and GPT2 for bioR dataset). The first tasks are retrieval, classification, comparison, inverse search. The conclusion is that LLMs should be accompanied by a knowledge base for inference.

**Strengths:**

The authors developed synthetic biography datasets and used them to train GPT2-X models, and vividly demonstrated the limitations of their trained model. The datasets, along with the four tasks, are very interesting and useful for continued research.

**Weaknesses:**

It is not a new finding that LLMs are not good at reasoning. The methodology that the authors used is not striking enough to serve as the last nail in the coffin of LLMs. Instead, the datasets and the tasks will encourage new training methods to improve the performances of LLMs or to motivate neural-symbolic systems, e.g. GPT2 + a simple symbolic reasoning system will be enough to promote the overall performance, and a knowledge base will not be necessary. Because knowledge bases themselves can be incomplete and inconsistent. This work did not explain why LLMs have such limitations, but provides a vague solution -- for knowledge manipulation tasks, LLMs should be accompanied by knowledge bases.  An ideal method (to prove that LLMs can not reason) shall explain why they cannot reason.

**Questions:**

1. In the training data of LLMs and knowledge bases, it is normal the one name can refer to more than one people, or objects. In your experiment, do you assume one name refers to one instance?

2. Following my comments above, is it easier to company LLMs with a simple symbolic reasoner to get better performances in the comparison and the classification tasks? The performance of inverse search can be solved by inverse the masks in the training stage (similar to the dual process of predicting words from the contexts, and predicting the contexts from words). This only doubles the training time.

---

### Official Review · Reviewer_otDf · 2023-11-01

**Soundness:** 3 good
**Presentation:** 3 good
**Contribution:** 3 good
**Rating:** 6
**Confidence:** 4

**Summary:**

This paper explores the ability of knowledge manipulation of language models based on GPT2 models, defining four tasks --- knowledge retrieval, knowledge classification, knowledge comparison, and knowledge inverse search. While these tasks are seen as a simple manipulation of knowledge, the language models are failed to successfully deal with these works, without making a reasoning step from an originally given form, unless either CoT is applied or the knowledge augmentation (in a manner of using permuted data and reversed data, etc.) is properly employed. Furthermore, the analysis is connected to GPT4 on some of the tasks including classification, comparison and inverse search, showing that even GPT4 suffers from handling those types of tasks, which are similarly designed to ones used for GPT2 models.

**Strengths:**

- The analysis done in this work, revealing that the language models are weak to directly handle simple types of knowledge manipulation unless CoT or knowledge augmentation is used, is unique and important, particularly showing that even LLM such as GPT4 suffers from the limitations on handling these tasks. This analysis is timely valuable and important, motivating other researchers to resolve the reported limitation in knowledge manipulation.
- The synthetic datasets used for analyzing the knowledge manipulation are prepared well and quite valuable as the standard setting for other researcher to investigate the issues.

**Weaknesses:**

- The language models used in this work are only GPT2, without using other open LLM such as llama and llama2. To convincingly confirm the conclusion, the evidences from other larger language models on the same dataset need to be provided.
- CoT is only limitedly applied to some tasks. But, CoTs could be applicable to retrieval and inverse search tasks. Both knowledge augmentation and CoT need to be applied to all tasks, but it is like the selective application depending on the tasks.
- While GPT4 is indirectly analyzed, the work doesn’t provide extensive experiments among various language models with different parameter sizes. Thus, although the paper mention that these limitations may not be resolved by scaling up, the experiment of language models across various parameters need to fully compared, to examine the extent of these limitations varying the parameter size.
- Only size attributes from the biography dataset are examined. Extension and generalization to other types of attributes and entities need to be explored.

**Questions:**

- Why CoT and knowledge augmentation are not applied to all the tasks fairly? Is there any reason for that CoT is NOT applied to retrieval and inverse search tasks.
- The experiments are done only using GPT2, without comparing with various language models with different parameter sizes. Working on the comparison across various parameter sizes is not necessary?
- The connection to GPT4 is interesting. But, why knowledge retrieval session doesn’t include the connection to GPT4?

---

### Official Review · Reviewer_BNdJ · 2023-11-06

**Soundness:** 4 excellent
**Presentation:** 4 excellent
**Contribution:** 3 good
**Rating:** 8
**Confidence:** 4

**Summary:**

This paper investigates pre-trained language models' knowledge manipulation abilities during inference, focusing on four types of manipulation and assessing models such as GPT-2, GPT-3, and GPT-4. The study reveals that these models struggle to manipulate knowledge, particularly in simple tasks like determining a person's birth month unless the same person's data is present in their training set. Training the model in a Chain of Thought (CoT) manner can improve this, but they still fail at inverse knowledge search. Even large models like GPT-4 face challenges in these tasks, indicating that these limitations may be intrinsic to generative language models and necessitate innovative techniques for enhancement. The paper's primary contribution is a synthetic dataset and the controlled experiments that confirms these inherent weaknesses.

**Strengths:**

- This work studies an important problem - knowledge manipulation which draw relevant interests towards important tasks like retrieval.
- The paper's methodology, datasets etc. are clear and well-organized.
- The paper's findings reveal inherent limitations in pre-trained language models' knowledge manipulation during inference, bearing substantial implications for future language model development and natural language processing system design. The synthetic dataset introduced also serves as a valuable resource for further research in this field.

Overall I enjoy reading the paper. While the conclusions are mainly drawn from the created synthetcic biography datasets, the studies on four different perspectives are relatively systematic.

**Weaknesses:**

I don't have much complaints about this work. The only thing is that I think it would be more solid for the paper to provide a section or a paragraph of discussion about the directions to improve language models on these failure modes.

**Questions:**

/